# A two-step procedure to generate utilities for the Infant health-related Quality of life Instrument (IQI)

Paul F. M. Krabbe[1]*, Ruslan Jabrayilov[1], Patrick Detzel[2], Livia Dainelli[2], Karin M. Vermeulen[1], Antoinette D. I. van Asselt[1]

**1** Department of Epidemiology, University Medical Center Groningen, University of Groningen, Groningen, the Netherlands, **2** Nestlé Research Center, Lausanne, Switzerland

* p.f.m.krabbe@umcg.nl

**Data Availability Statement:** All relevant data are within the paper and its Supporting Information files.

## Abstract

### Background

Because of a lack of preference-based health-related quality of life (HRQoL) instruments suitable for infants aged 0–12 months, we previously developed the Infant QoL Instrument (IQI). The present study aimed to generate an algorithm to estimate utilities for the IQI.

### Methods

Via an online survey, respondents from the general population and primary caregivers from China-Hong Kong, the UK, and the USA were presented 10 discrete choice scenarios based on the IQI classification system. An additional sample of respondents from the general population were also asked if they considered the examined health states to be worse than death. Coefficients for the IQI item levels were obtained with a conditional logit model based on the responses of the primary caregivers for IQI states only. These coefficients were then normalized using the rank-ordered logit model based on the responses from the general population who assessed "death" as a choice option. In this way, the values were rescaled from full health (1.0) to death (0.0), and consequently, they became suitable for the computation of quality-adjusted life years.

### Results

The total sample consisted of 1409 members of the general population and 1229 primary caregivers. Results indicated that, out of the 7 IQI items ("sleeping," "feeding," "breathing," "stooling/poo," "mood," "skin," and "interaction"), "breathing" had the highest impact on the HRQoL of infants. Moreover, except for "stooling," all item levels were statistically significant. The general population sample considered none of the health states as worse than death. The utility value for the worst health state was 0.015 (State 4444444).

**Funding:** This study has been funded by Nestec Ltd to finance the research activities of RJ, KV, AvA and PK. PD and LD are employed at Nestlé Research Center. The funder provided support in the form of salaries for authors [RJ, KV, PD, LD, AvA, PK], but did not have any additional role in the study design, data collection and analysis, decision to publish, or preparation of the manuscript. The specific roles of these authors are articulated in the 'author contributions' section.

**Competing interests:** As an extension of the health outcome instrument and mobile application presented in this paper, additional measurement models, tools and instruments are developed by PK as part of academic/commercial activities. This does not alter our adherence to Plos One policies on sharing data and materials.

## Conclusions

The IQI is the first generic instrument to assess overall HRQoL in 0–1-year-old infants by providing values and utilities. Using discrete choice experiments, we demonstrated that it is possible to derive utilities of infant health states. The next step will be to collect IQI values in a clinical population of infants and to compare these values with those of other instruments.

## Introduction

Regulatory authorities and governmental organizations generally require studies to evaluate the value of health interventions. Many of these bodies recommend using a summary measure of health outcome, such as quality-adjusted life years (QALYs), as the unit of health benefit in economic evaluations [1]. Central to the computation of QALYs is the "quality" component, which is mostly quantified in terms of concepts such as health-status or health-related quality of life (HRQoL) [2]. Such HRQoL measures suited for QALYs are expressed in a single metric for health states or conditions, anchored on a unidimensional scale. They are often classified as "generic" and "preference-based." While "generic" means that such tools can be applied across a wide range of populations and interventions, therefore allowing comparisons among them, "preference-based" means that these measurement methods are used to arrive at values that place health states on a scale. These methods explicitly incorporate weights that reflect the importance attached to specific health items (also known as attributes, domains, dimensions, or indicators) [3,4,5]. Preference-based methods stemming from economics, such as the standard gamble and time trade-off, are constructed such that they directly produce values on a scale, where 0.0 is equal to death and 1.0 is full health, and these values can be applied in QALY computations, where they are called utilities. Other types of preference-based methods require extensions or additional exercises to normalize values because death does not appear on the scale.

To overcome the lack of generic preference-based HRQoL instruments suitable for infants aged 0–12 months, the Infant Quality of life Instrument (IQI) was developed recently [6]. For the selection of the relevant domains (items) for the IQI, a multi-step development process began by extracting candidate health concepts from relevant measures that were identified by searching the literature. Next, panels, with experts from Asia, Europe, New Zealand, and the United States of America (USA), and two surveys, with primary caregivers in New Zealand, Singapore, and the United Kingdom (UK), evaluated the relevance of the candidate health concepts, organized them into attributes based on their similarities, explored alternative attributes, and generated response scales. Additional interviews assessed the cross-cultural interpretability, parents' understanding of health attributes, and the usability of the mobile application.

We also conducted a study in which the IQI was used in a preference-based method (discrete choice) to generate values [7]. However, in the present study, we went one step further to generate utilities that are applicable in the computation of QALYs. It is unclear which method is most appropriate to obtain values and utilities for adolescents or children. Several studies discourage the use of conventional economic valuation methods, such as the time trade-off, with this age group because caregivers (proxies) are apparently unwilling to trade off a child's life years, leading to relatively high values for poor health states [8,9,10]. In addition, these methods are not only complex but are also associated with numerous theoretical violations, biases, and practical problems [11, 12]. Deriving utilities from the general public who

represent the societal perspective, is the prevailing approach in economic evaluations because the general public, being taxpayers and potential users of the healthcare system, are considered the most reasonable assessors [13].

The present study aimed to explain how utilities for the IQI health states were generated with a novel two-step choice-based modeling procedure that helps locate the position of death on the IQI scale [14]. Specifically, this process involved the following two steps: 1) deriving values for a set of IQI health states from primary caregivers of infants based on a discrete choice modeling exercise, 2) normalizing the values obtained in Step 1 to an anchored 0.0–1.0 scale using utilities derived from responses collected from a general population sample.

## Methods

### Instrument

The IQI includes 7 health items ("sleeping," "feeding," "breathing," "stooling/poo," "mood," "skin," and "interaction"), each relevant at each time point up to 1 year of age [5]. Each item consists of 4 levels, most of which are ranked by severity. For instance, the levels for "sleeping" are 1: sleeps well, 2: slightly affected sleep, 3: moderately affected sleep, and 4: severely disturbed sleep. Only for the item "skin," the levels phrased qualitatively rather than quantitatively. The IQI can be administered through a mobile application (www.chateau-sante.com/iqi); its usability was previously tested on primary caregivers [5] and further improved in light of their opinions (Fig 1). For each health item, primary caregivers can select the level that best applies to their infant. In this way they "construct" an IQI health state that forms an overall health description that is expressed in 7 digits, e.g., 3231421, which would equate "moderately affected sleep, slight feeding problems, moderate breathing problems, normal stool/poo, inconsolable crying, dry or red skin, highly playful/highly interactive".

### Recruitment of respondents

Primary caregivers of infants and toddlers (0–3 years old) and people from the general population were recruited in China-Hong Kong, the UK, and the USA to conduct the main study. These countries were selected for practical reasons, since they are culturally different yet share one language, thus eliminating the need for translation at this phase, and enabling the analysis of possible cross-cultural differences in the results, and therefore improving generalizability. Clear instructions were given to all participants. While the instrument targets infants up to one year, in the survey we chose to include primary caregivers of 2- and 3-year-olds as well, to enable the recruitment of a larger sample. We assumed that the caregivers could recollect their experiences of the first year of their infant's life quite easily. The general population sample included both parents (of children with varying ages) and respondents without children, to be as representative as possible. The latter were interviewed because they might think differently about the value of life in different health conditions.

In an additional study, a separate smaller sample was drawn in the USA from among members of the general population. This study was conducted to gain a better understanding of the severe IQI health states to explore and confirm our findings from the main study.

All respondents were contacted through a market research company (Survey Sampling International, SSI). Respondents who completed the entire survey received a small financial compensation from SSI. The rewards were defined by the company's (SSI) internal agreements with the groups of respondents. The Medical Ethics Review Committee at the University Medical Center of Groningen issued a waiver for this study, indicating that the pertinent Dutch legislation (the Medical Research Involving Human Subjects Act) did not apply for this non-interventional study (METc2017.115).

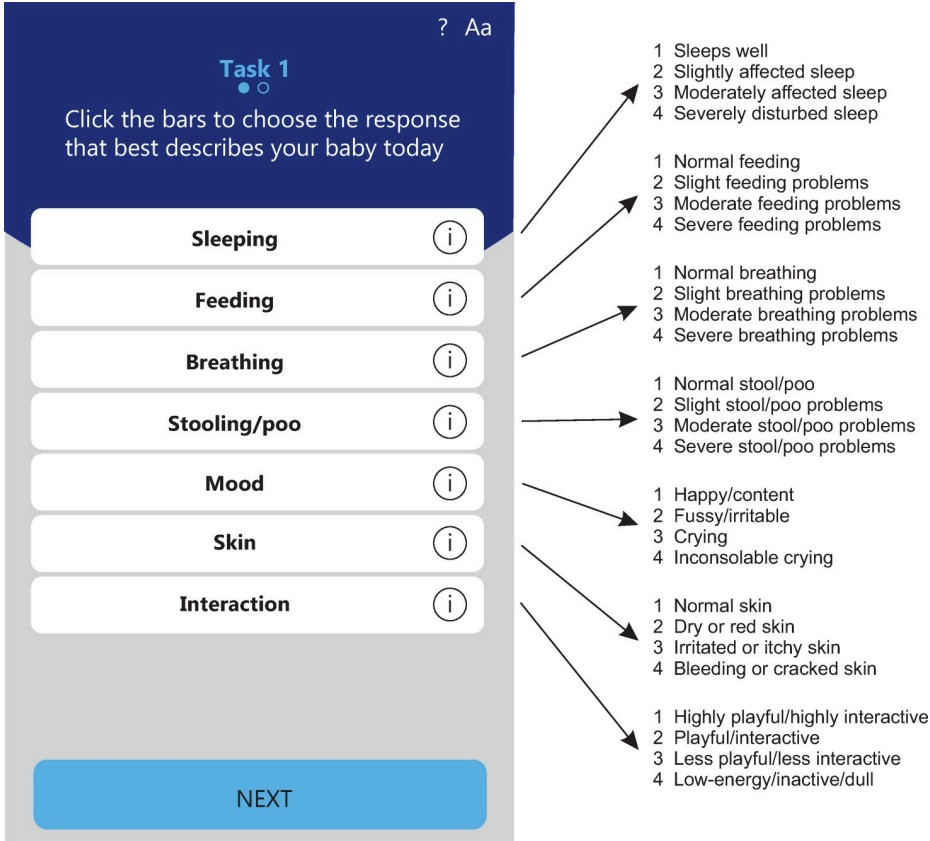

**Fig 1. Infant Quality of life Instrument (IQI) health items and their levels (left: screenshot of the mobile application for the IQI).**

## Response task

Respondents in the main study were presented with 10 discrete choice scenarios in an online survey. They were requested to indicate which of the two hypothetical health states presented was better (Fig 2). The order of the items (e.g., sleeping, breathing, interaction) was randomized for every respondent. Prior to each paired comparison task, respondents were instructed about two assumptions- that the health states presented in the task would occur in the first year of life, and that it was uncertain as to what would happen after that year. In adult populations, the timespan of the comparison typically ends at death. As this does not seem appropriate for a younger population, we believe that by only describing the situation in the first year and not being explicit about what comes after, the focus of the comparison would be on the first year, as intended. After the paired comparison task, respondents were asked to indicate whether they considered any of the two health states as being worse than death.

## Discrete choice design

With the IQI classification system, a total of $4^7$ (16,384; 7 items with 4 levels) health state classifications are possible. Consequently, 134,209,536 ([[16,384 * 16,384] - 16,384]/2) unique IQI health state pairs can be devised. To select a smaller number of pairs from this large pool, several criteria were used. The first was based on the fact that comparisons containing a dominant health state, i.e., a state with all items at a better level than the comparator state (e.g., 222222 vs.

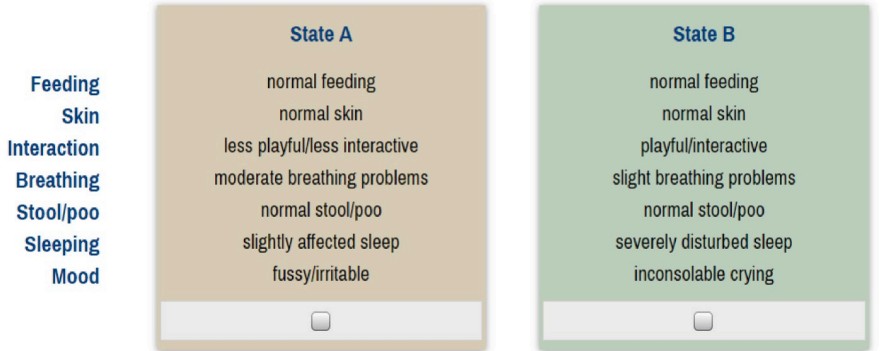

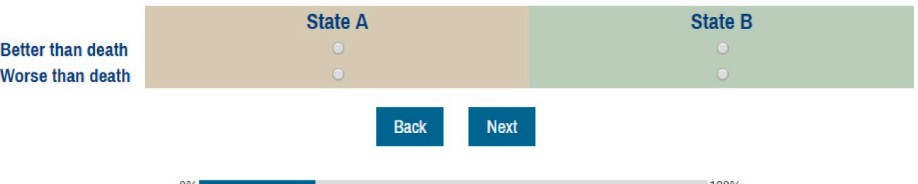

**Fig 2. Screenshot of the discrete choice task and the complementary "better or worse than death" task.**

3333333), are not ideal for selection, as they do not add information. Therefore, pairs with one health state dominating the other were excluded from the task. The second criterion was the selection of pairs with a certain overlap, deemed to facilitate comprehension. Specifically, we included pairs that varied on 4 items and overlapped on 3 (see Fig 2: only 4 of the items vary between Infant A and B). In the 4 items that varied, 2 represented better-off item levels in Alternative A than Alternative B, and 2 represented worse-off item levels in Alternative A than in Alternative B. The third criterion was that, in at least half of the tasks, the maximum difference in item levels between the health states was set to 1. For example, Level 2 could be compared to Level 1 and 3, but not to Level 4. The rationale for restricting the range was to avoid comparisons of health states that were very different from each other. However, the remaining set of tasks would allow larger differences. The above criteria of our study design were all programmed in MATLAB [15].

A discrete choice design generally entails selecting items from the full set of all possible health states. Accordingly, the worst health states at the top and bottom of the scale are often absent. The included health states are often "mild" and they are thus likely to be considered better than "death." Therefore, to gain insight into the worst health states as well as their relationship to "death," we conducted an additional discrete choice study. In this study, half of the participants were asked to compare two IQI health states composed of the worst item levels (i.e., severe problems) except for one item with Level 3 (S1 Fig). In total, $7 \times 6 = 42/2 = 21$ such health pairs are possible. Then, the other half were asked to compare these worst health states

(6 items with Level 4 and 1 item with Level 3) to the health state "death" (S2 Fig). In total, 7 such comparisons are possible.

## Analyses

Coefficients for the IQI item levels were obtained with a conditional logit model (Stata, clogit) based on the responses of the primary caregivers (the same analysis was also conducted for the general population but was not a part of our strategy to arrive at utilities). Initially, the first level (i.e., no problems) of each health item was taken as the reference category. The coefficients for the remaining 3 levels were then estimated using 21 dummy variables (7 × 3). After preliminary analyses to assess "interaction," the reference category was changed to the second level, because the first level in this health item did not represent the best health condition; i.e., the second level had the highest coefficient. This implies that the first level "highly playful/ highly interactive" would result in a lower score in the valuation than would the second level "playful/interactive."

The value of a health state $j$ for individual $i$ is denoted by $V_{ij}$. It is assumed that $V_{ij}$ is a linear combination of the levels on the health items plus an error term $\varepsilon_{ij}$ for the individual. The model specification is

$$V_{ij} = \sum_{j=1}^{n} \beta x_{ij} + \varepsilon_{ij} \qquad (1)$$

where $\beta$s represents a vector of 21 regression coefficients and $x_{ij}$ a vector of 21 binary dummy explanatory variables ($x^{\delta\lambda}$), where $\lambda$ indicates the levels of each of the 7 items ($\delta = 1, 2, \ldots, 7$) for a health state [16]. For example, $x^{42}$ represents the second level (*slight problems*) of the fourth item ("breathing"). All computations and the visualization of the results were carried out using Stata 15.0 [17], R programming language [18], and SigmaPlot 14.0 [19].

Next, the estimated coefficients from the primary caregivers were normalized. By normalizing we mean that the values are transformed or rescaled to produce a common utility scale (0–1). For this purpose, data obtained from the "death" task (Fig 2) performed by the general population sample (the same analysis was also conducted on the data collected from the primary caregivers but it was not a part of our strategy to arrive at utilities) were analyzed by the rank-ordered logit model (Stata, rologit) [20].

$$V_{ij} = \sum_{j=1}^{n} \beta x_{ij} + \beta D + \varepsilon_{ij}$$

In addition to the 21 dummy variables, one for "death" ($D$) was used to rescale the values from 0.0 (death) to 1.0 (full health). By dividing the remaining coefficients by the estimated death coefficient, all health states were rescaled from 0.0 to 1.0. The utility of a health state is calculated as 1 minus the sum of coefficients for the corresponding levels on the health items.

## Results

### Respondents

In total 2638 respondents were recruited from China-Hong Kong (n = 818), the UK (n = 920) and the USA (n = 890). The sample consisted of members of the general population (n = 1409) and primary caregivers of infants (n = 1229). The average age of the respondents was 37 (median: 35) years, with 73% of the sample comprising females (Table 1). For the subsample of primary caregivers of infants, the average age was lower (33 years) and the proportion of females (93%) was substantially higher, which was expected given that a primary caregiver

**Table 1. Demographics of the various study samples.**

| Demographic characteristics | General population (N = 1409) | Primary caregivers (N = 1229) | Total sample (N = 2638) | Additional sample (n = 1027) |
|---|---|---|---|---|
| Country | | | | |
| China-Hong Kong | 421 (30%) | 407 (33%) | 828 (31%) | - |
| UK | 516 (37%) | 404 (33%) | 920 (35%) | - |
| USA | 472 (33%) | 418 (34%) | 890 (34%) | 1027 (100%) |
| Gender | | | | |
| Male | 632 (45%) | 81 (7%) | 713 (27%) | 523 (51%) |
| Female | 777 (55%) | 1148 (93%) | 1925 (73%) | 504 (49%) |
| Age (years) | | | | |
| Mean | 41 | 33 | 37 | 32 |
| Median | 39 | 33 | 35 | 33 |

would typically be a young mother. For the additional study, a total of 1027 respondents (N = 523 for Task 1, N = 504 for Task 2) were recruited from among the members of the general population in the USA; 49% of the respondents were females and the mean age was 32 (median: 33) years.

## States worse than death

Among the health states in the main study, the one mentioned most frequently as being worse than death was 4241241 (2.2%) for the general population and 4244231 (2.4%) for the primary caregivers. Among the 7 worst health states that were included in the additional study for the general population sample only, the percentage of respondents indicating a health state as being worse than death ranged between 20% and 25%.

## Coefficients IQI items

The coefficients for the levels of the 7 IQI items based on judgments made by the primary caregivers showed that "breathing" had the highest impact on the HRQoL of infants (Table 2). For the sample of primary caregivers, all levels of the 7 IQI items proved to be statistically significant (coefficient > 0.0), except "stooling" Level 3 (moderate stool/poo problems). Coefficients were negative for most of the levels of these items and followed a logical order (i.e., slight problems < moderate problems < severe problems). Negative coefficients implied that a particular level was worse than the baseline, which in our study was the first level of each health item, except for the interaction item. Moreover, the less preferable an item was considered, the higher its coefficient was, in the negative direction. For 4 items, i.e., "stooling," "mood," "skin," and "interaction," the order of the coefficients was not strictly monotonously decreasing. For example, the coefficient (not statistically significant) for the third level (moderate problems) had a positive coefficient for "stooling" in the overall sample, indicating that it was more preferable than the baseline level (no problems) and also than the second level (slight problems). For a more detailed discussion about the coefficients, see our earlier publication [6]. The coefficients obtained from the primary caregivers and the general population based on the choice experiment without "death" were rather comparable (Table 2, S3 Table). Slightly different results were obtained in the more complex choice experiment, which included the "death" options. Specifically, 4 non-statistically significant coefficients were observed for the general population sample and 5 were observed for the caregivers. The coefficients within the item "interaction" were almost equal for the general population sample, as well as the "death" coefficient (Table 2, S1 Table).

**Table 2. Parameter estimates for the levels of the 7 IQI health items separately for the primary caregivers, the general population, and the normalized primary caregivers' parameters.**

| | Primary caregivers (DC) | | | General population (DC + Death) | | | Primary caregivers normalized scale (utilities) |
| --- | --- | --- | --- | --- | --- | --- | --- |
| | Coefficient | SE | Significance | Coefficient | SE | Significance | Coefficient |
| Sleeping (2) | -.246 | .046 | .000 | -.192 | .036 | .000 | -.056 |
| Sleeping (3) | -.403 | .052 | .000 | -.226 | .036 | .000 | -.092 |
| Sleeping (4) | -.774 | .052 | .000 | -.599 | .036 | .000 | -.176 |
| Feeding (2) | -.158 | .046 | .001 | -.191 | .035 | .000 | -.036 |
| Feeding (3) | -.162 | .050 | .001 | -.143 | .035 | .000 | -.037 |
| Feeding (4) | -.683 | .054 | .000 | -.381 | .035 | .000 | -.155 |
| Breathing (2) | -.395 | .049 | .000 | -.121 | .036 | .001 | -.090 |
| Breathing (3) | -.585 | .052 | .000 | -.204 | .036 | .000 | -.133 |
| Breathing (4) | -1.047 | .055 | .000 | -.664 | .036 | .000 | -.238 |
| Stooling (2) | -.100 | .045 | .003 | .037 | .033 | .272* | -.023 |
| Stooling (3) | -.039 | .052 | .455* | .178 | .035 | .000 | -.009 |
| Stooling (4) | -.268 | .063 | .000 | -.058 | .038 | .130* | -.061 |
| Mood (2) | -.509 | .047 | .000 | -.303 | .034 | .000 | -.116 |
| Mood (3) | -.380 | .049 | .000 | -.219 | .035 | .000 | -.086 |
| Mood (4) | -.613 | .058 | .000 | -.300 | .038 | .000 | -.139 |
| Skin (2) | -.166 | .045 | .000 | -.026 | .034 | .459* | -.038 |
| Skin (3) | -.120 | .048 | .014 | -.026 | .034 | .447* | -.027 |
| Skin (4) | -.416 | .056 | .000 | -.195 | .035 | .000 | -.095 |
| Interaction (1) | -.170 | .047 | .000 | -.081 | .035 | .020 | -.039 |
| Interaction (3) | -.360 | .049 | .000 | -.073 | .036 | .043 | -.082 |
| Interaction (4) | -.531 | .052 | .000 | -.078 | .037 | .035 | -.121 |
| Death | N. A. | N. A. | N. A. | -2.307 | .075 | .000 | N. A. |

* Coefficients not significantly different than the baseline category (Level 1)

IQI = Infant Quality of life Instrument

Some differences were observed between the different countries. In China, for example, sleeping was considered the most important item, whereas it was less important in the UK and USA. Feeding was considered more important in the USA, while breathing was considered less important in China. The coefficients for "death" were -1.769, -2.827, and -2.474, respectively, for China, the UK, and the USA. For more details see [7, S2 Table].

## Utilities for IQI health states

According to the general population sample, no health state had a value below 0.0 (i.e., none was considered worse than death), as indicated by the normalized health state values (utilities). Values of the primary caregivers were normalized by the IQI utilities derived from the general population. The worst health state among the primary caregivers was 4444444, with a value of -4.332. Among the general population, the utility for that state was 0.015. The distance between 0.015 and zero (death) was used to normalize the caregivers' values into utilities (S3 Fig). The final coefficients that were applied to compute utilities for IQI health states are presented in the last column of Table 2. Utilities for all possible IQI health states (N = 16,384) were calculated by subtracting from 1.0, the coefficients corresponding to the levels for the 7 IQI items (Fig 3). In addition, the distribution of utilities for all IQI states were calculated (Fig 4). The utility for the worst IQI state (4444444) was 0.015, that for State 3333333 it was 0.534, and that for State 2222222 it was 0.641. The best IQI state (utility score 1.000) was 1111112, since Level

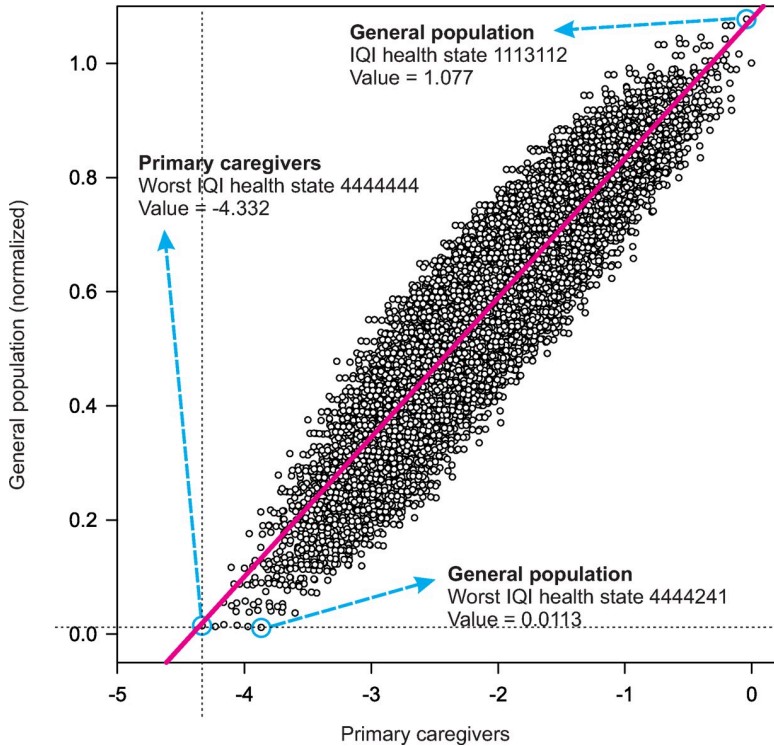

**Fig 3. Estimated health-state values obtained for the primary caregivers with discrete choice tasks (pairs of IQI states) and for the general population with a ranking task (two IQI states + death).**

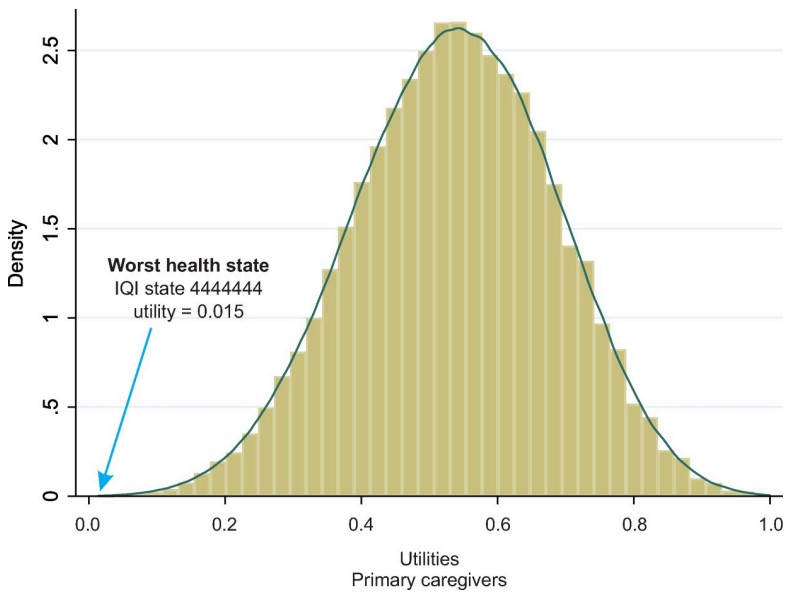

**Fig 4. Estimated health-state utilities obtained for the primary caregivers (discrete choice task) normalized for the location of death (based on values derived from task with death by the general population).**

2 was the reference level for the interaction domain, and therefore, 1111112 was considered to represent perfect health. State 1111111 had a utility score of 0.961 (a complete list of utilities is available on request).

The same analysis for the three countries separately showed that the utilities for IQI state 4444444 were -0.121, 0.114, and -0.022, respectively, for China, the UK, and the USA. Distributions for the three countries showed that the distribution of utilities of the IQI health states for China is different in comparison with that in the UK and USA (S4 Fig). In the Chinese sample, health states were considered more worse and a small proportion of states were considered worse than death. Normalized coefficients for each country have been presented in S3 Table.

## Discussion

In this study, we explained a novel two-step procedure to generate utilities for the IQI, relying on a sample comprising both primary caregivers and members of the general population. Values were derived for IQI health states based on responses from primary caregivers (e.g., proxies). Results indicated that out of the 7 IQI items, "breathing" had the highest impact on the HRQoL of infants. Moreover, except for "stooling," all item levels were statistically significant. Subsequently, these values of the caregivers were normalized into utilities by using information on the location of death on the scale, derived from a general population sample. Findings revealed that none of the health states contained in the IQI was worse than death.

According to the current convention in QALY computation, the lower anchor on the scale should be 0.0, defined as a state equivalent to death [21,22,23]. However, a critical problem that is associated with current economic valuation methods is the lack of a reliable method to determine the position of "death" on the HRQoL scale. The use of the concept of "death" in health measurement methods is a controversial issue [24]; some find it astonishing that health economists make death a central element of their valuation methods [25]. In the present study, the lowest utility for the worst IQI health state (4444444) was only just better than death, namely 0.015. A widely applied generic preference-based health status instrument like the EQ-5D has presented, in its original 3-level version (EQ-5D-3L), a negative value of -0.59 for its worst health state [26]. However, in the new 5-level version (EQ-5D-5L) the value for its worst state is less negative (-0.29), as observed in many other preference-based competitor instruments [27]. The Health Utility Index Mark III, for example, shows a value of -0.36 for its worst state, but other widely used instruments, such as the Quality of Wellbeing index, 15D, SF-6D, and AQol-8D, show positive values of 0.32, 0.11, 0.20, and >0.20, respectively [28,29]. Additionally, in practice, since reimbursement decisions will be made based on a broader range of criteria than just cost-effectiveness, we do not expect that having the lowest IQI value higher than 0 would be a disadvantage for the infant population.

In this study, we used conventional discrete choice models to derive utilities for the IQI. Recently, we introduced a novel and straightforward measurement system to derive weights (coefficients) to estimate values and utilities. This measurement system is based on the item response theory and discrete choice methods [30]. This model continuously collects responses from patients or proxies and integrates findings from the health classification and preference tasks [31,32]. With this new method, the health-outcome measures collected are not only preference-based but also patient-centered [33,34,35]. Once tested and validated, we intend to apply this measurement model in future clinical IQI studies to estimate utilities.

The country-specific results for the primary caregivers revealed a number of differences. Overall, the UK and USA appeared more alike, while China was slightly different. This could have its source in the different cultures of the countries in the sample, but given that sample

sizes of around 400 were used for each country, a part of this finding may also be an artefact. In China, sleep appeared to be more important, whereas in the UK and USA, higher coefficients were observed for most other attributes, and more value was attached to breathing, mood, and interaction. Eventually, a larger sample should be used to determine final value sets based on country-specific preferences.

A limitation associated with this study is that no obvious state worse than death was found. From a statistical perspective, values are likely to be biased when a significant proportion of the sample takes the normative position that all life is worth living [36]. However, the additional analysis in which 21 pairs of severe health states were utilized demonstrated ample effect on the estimation of the worse health states. The same holds for the analysis that included death. Therefore, the utility (0.015) of the worst IQI health state (4444444) seems credible as it was confirmed by the statistical models and by the simple preference task used in the additional analysis. Another limitation was that detailed characteristics of the respondents were not available, apart from country, age, and sex. Therefore, we were not able to, for instance, say which part of the general population sample comprised parents, nor could we perform stratified analyses based on socioeconomic status. It should be kept in mind that the present study was largely intended as a proof of principle to demonstrate how the process of generating normalized values (utilities) can take place and to provide a first value set for the IQI. From that perspective, the generalizability of the results is of less importance at this stage.

The IQI is the first generic preference-based instrument to assess overall HRQoL in 0–1-year-old infants. We demonstrated that it is possible to generate an algorithm to derive utilities from the infant health states using discrete choice experiments. The next step will be to use the IQI in a clinical population of infants to see how it performs as compared to other instruments, and to collect responses to refine the value set over time.

## Supporting information

**S1 Fig. Additional study (Part 1) in which the 7 worst IQI health states (apart from 4444444) were assessed against each other.**
(EPS)

**S2 Fig. Additional study (Part 2) in which the 7 worst IQI health states (apart from 4444444) were assessed against death.**
(EPS)

**S3 Fig. Relationship between the values for the primary caregivers (bottom) and the normalized values for the general population (top).**
(EPS)

**S4 Fig. Normalized values (utilities) per country (same X- and Y-axis).**
(EPS)

**S1 Table. Parameter estimates for the levels of the 7 IQI health items separately for the general population and primary caregivers.**
(DOCX)

**S2 Table. Parameter estimates (values) for the levels of the 7 IQI health items for the primary caregivers, per country.**
(DOCX)

**S3 Table. Parameter estimates (normalized scale = utilities) for the levels of the 7 IQI health items for the primary caregivers, per country.**
(DOCX)

## Author Contributions

**Conceptualization:** Paul F. M. Krabbe.

**Data curation:** Ruslan Jabrayilov.

**Formal analysis:** Ruslan Jabrayilov.

**Funding acquisition:** Paul F. M. Krabbe.

**Investigation:** Paul F. M. Krabbe, Ruslan Jabrayilov, Patrick Detzel, Livia Dainelli, Karin M. Vermeulen, Antoinette D. I. van Asselt.

**Methodology:** Paul F. M. Krabbe, Ruslan Jabrayilov, Patrick Detzel, Livia Dainelli.

**Project administration:** Paul F. M. Krabbe, Patrick Detzel, Livia Dainelli, Karin M. Vermeulen, Antoinette D. I. van Asselt.

**Software:** Paul F. M. Krabbe, Ruslan Jabrayilov.

**Supervision:** Paul F. M. Krabbe.

**Validation:** Paul F. M. Krabbe, Ruslan Jabrayilov, Patrick Detzel, Livia Dainelli, Karin M. Vermeulen, Antoinette D. I. van Asselt.

**Visualization:** Paul F. M. Krabbe, Ruslan Jabrayilov.

**Writing – original draft:** Paul F. M. Krabbe, Ruslan Jabrayilov, Patrick Detzel, Livia Dainelli, Karin M. Vermeulen, Antoinette D. I. van Asselt.

**Writing – review & editing:** Paul F. M. Krabbe, Ruslan Jabrayilov, Patrick Detzel, Livia Dainelli, Karin M. Vermeulen, Antoinette D. I. van Asselt.

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
