## [Decision Letter · Decision Letter 0]

30 Oct 2019

PONE-D-19-19703

A two-step procedure to generate utilities for the Infant health-related Quality of life Instrument (IQI)

PLOS ONE

Dear Mr. Krabbe,

Thank you for submitting your manuscript to PLOS ONE. After careful consideration, we feel that it has merit but does not fully meet PLOS ONE’s publication criteria as it currently stands. Therefore, we invite you to submit a revised version of the manuscript that addresses the points raised during the review process.

We would appreciate receiving your revised manuscript by Dec 13 2019 11:59PM. To enhance the reproducibility of your results, we recommend that if applicable you deposit your laboratory protocols in protocols.io, where a protocol can be assigned its own identifier (DOI) such that it can be cited independently in the future. For instructions see: http://journals.plos.org/plosone/s/submission-guidelines#loc-laboratory-protocols

We look forward to receiving your revised manuscript.

Kind regards,

Jing Tian

Academic Editor

PLOS ONE

Journal Requirements:

'This study has been funded by Nestec Ltd to finance the research activities of RJ, KV, AvA and PK. PD and LD are employed at Nestlé Research Center.

The funder provided support in the form of salaries for authors [RJ, KV, PD, LD, AvA, PK], but did not have any additional role in the study design, data collection and analysis, decision to publish, or preparation of the manuscript.'

We note that one or more of the authors are employed by a commercial company: Nestle

1. Please provide an updated Competing Interests Statement declaring this commercial affiliation along with any other relevant declarations relating to employment, consultancy, patents, products in development, or marketed products, etc. 

Please include an updated Competing Interests Statement in your cover letter. We will change the online submission form on your behalf.

Additional Editor Comments (if provided):

Reviewers' comments:

Reviewer's Responses to Questions

**Comments to the Author**

1. Is the manuscript technically sound, and do the data support the conclusions?

Reviewer #1: Yes

Reviewer #2: Partly

Reviewer #3: Partly

2. Has the statistical analysis been performed appropriately and rigorously? 

Reviewer #1: Yes

Reviewer #2: Yes

Reviewer #3: Yes

3. Have the authors made all data underlying the findings in their manuscript fully available?

Reviewer #1: Yes

Reviewer #2: No

Reviewer #3: Yes

4. Is the manuscript presented in an intelligible fashion and written in standard English?

Reviewer #1: Yes

Reviewer #2: Yes

Reviewer #3: Yes

5. Review Comments to the Author

Reviewer #1: The purpose of this study is to generate an algorithm to estimate utilities for the infant quality of life instrument (IQI) using a two-step choice-based modelling approach. The approach and methods used for the study seem appropriate and I have just a few comments.

This study uses a population from the UK, USA and China-Hongkong and my question relates to why this population was used. Was it based on availability or was it to ensure generalisability? In addition, I am not entirely sure why the second study was limited to USA. Perhaps the authors need to comment on the generalisability of the results.

It would help if a table with the utility values for the various health states is presented in the appendix. Linked to this point, the lowest/worst health state is represented by 0.015 and as a result, there is a gap between this value and zero which is usually considered the anchor point for utility scales. As a result of this, the use of the IQI would imply that there could potentially be an over-estimation of utility for the severest health states which could potentially lead to a situation where the paediatric population would be disadvantaged compared to e.g. an adult population. Perhaps, the authors should also comment on the implications of using this instrument to assess the cost-effectiveness of interventions.

Reviewer #2: General comments to the authors

Thank you to the authors for providing me with the opportunity to review this novel and interesting quality of life research paper. The authors identified a substantial gap in the literature in their previous published work and are now developing the algorithm for the previously established descriptive system of the IQI to capture and assess proxy-reported outcomes for the derivation of health state utility valuations for infants aged 0-1 years.

I note that the underpinning theory regarding the development of the IQI’s descriptive/classification system by Krabbe et al’s authorship team in 2018 (https://doi.org/10.1371/journal.pone.0203276) where the views of the primary caregivers serve as the proxy and provide the response. This paper is the logical next step to that published work.

Overall, I consider that it would be useful to provide some additional background in the Introduction section of the manuscript regarding the development of the IQI’s descriptive/classification system, including the literature search and use of expert panels in this development. This would contextualise the reasons for this paper to the broader readership.

I also suggest that the Discussion section could be more circumspect about the development of this initial (albeit important) value set. Please include caveats regarding future confirmatory and clinical studies and perhaps refinement of the value set over time.

Finally, I also suggest that the manuscript should be thoroughly checked for grammatical errors.

I recommend the paper for publication if my key concerns are appropriately addressed as outlined below.

Specific comments to the authors:

Introduction section:

Line 69: The introductory paragraph could perhaps mention full health economic evaluation and cost-utility analyses to contextualise the generation of utilities as a health economic input metric for cost-utility analyses (refer Drummond et al 2015).

Line 86: As noted in the general comments, please include some additional background regarding the development of the IQI’s descriptive system.

Line 93 Aims: The ‘aims’ paragraph could be improved by describing how you planned to achieve your key objective with some further detail and perhaps numbered sequentially.

Methods section:

Line 108: please provide further explanation regarding the ‘skin’ dimension being classified ‘qualitatively’. Is the ‘mood’ dimension perhaps phrased qualitatively too? Do you mean that there needs to be a more subjective assessment of the response?

Line 110: is the term ‘parents’ = primary carers? Are there ‘primary carers’ of the infants who are not classified as parents? Please clarify.

Line 219: Could you please relabel this “Recruitment of Participants (or Respondents)”. Please expand on this fundamental section: the recruitment strategy requires additional explanation . I note that a fulsome explanation was provided of the discrete choice design and the subsequent analyses. It is insufficient to say that participants were reached through a market research company. I need to understand the sampling and any bias. I also need to understand why participants from the USA were recruited only for the second study. Please also clarify in this section the meaning of the ‘main study’ and the ‘additional study’.

Results section:

I note that the line numbers on the PDF files were not reproduced from the Results section, therefore the below comments are outlined section by section.

Given the international nature of the cohort, stratified results should be reported beyond section 3.1 to highlight any cultural/country differences.

Section 3.4: Please provide further contextualisation regarding the utility derived for ‘1111112’ and ‘1111111’. The reference point and the derivation of a utility score of 1.0 for 1111112 and 0.96 for 1111111 may require further clarification for the broader readership.

Discussion section:

The Discussion is perhaps the weakest section of the paper and should be expanded to showcase the key and secondary findings of the paper, thoroughly outline all of the limitations of the study, and provide a robust conclusion.

The first paragraph should provide an overall summary of the findings. One sentence is inadequate and this sentence describes what was done rather than a succinct summary of the key findings of the paper.

The Discussion also does not outline the international nature of the cohort, nor issues surrounding cultural differences for the proxy-respondents as primary carers of infants and therefore the generaton of potentially different value sets?

In the second paragraph of the Discussion, perhaps remove the word ‘first’ – the authors do not then go to ‘second’, ‘third’ etc to expand on the key findings. The second paragraph could also reference the AQoL-8D multi-attribute utility instrument’s algorithmic range as an example of an instrument that does not record a utility value that is less than zero.

In the third paragraph, I would remove the statement that “It is likely that states worse than death are less self-evident than generally thought and that the lowest utility for the EQ-5D-3L may have been an accidental finding”. Perhaps the suggestion could be underpinned by statements about the instruments only 243 health states and that the instrument is relatively insensitive to complex and chronic disease states. Please use appropriate referencing for this statement.

The fourth paragraph could be tightened to provide some additional explanation and contextualisation regarding the underpinning model and the advantages of the model. This paragraph in its current form is somewhat jumbled.

Limitations section. Please be more circumspect in this section – one limitation only is outlined. The recruitment/sampling strategy is not properly described therefore I can not comment on limitations regarding the sampling strategy. Similarly, stratified results are not presented I would expect that cultural differences would be evident in stratified results. Stratified results could also be the subject of further discussion.

The conclusion should provide additional explanation regarding future research and confirmatory studies.

Reviewer #3: The sample consisted of members of the general population (n=1409) and infant caregivers (n=1229). Were the members of the general population parents? If not, is there reason to believe that the sample of the general population would answer the questions significantly differently than the infant caregivers? I’m assuming from some of the information that follows in the manuscript, that yes, the subsamples answer some questions differently, but

In section 3.2 “states worse than death”, the authors write, “Among the health states in the main study, the one most frequently mentioned as being worse than death was 4241241 (2.2%) for the general population and 4244231 (2.4%) for the primary caregivers.” The digits are poorly explained in section 2.1 instrument. A slightly more detailed explanation would be useful for the reader earlier in the paper (perhaps expand the example given in line 113, “e.g., 3231421 equates to moderately affected sleep, slight feeding problems, moderate breathing problems, sleeps well, inconsolable crying, dry or red skin, highly playful/highly interactive.” Because as it is, I’m not even certain that I have interpreted this correctly. Is this correct?

Figure 2 is confusing because at the top it says, “Suppose that an infant’s first year of life is spent mainly in either State A or State B and that its health is uncertain afterwards.” At the bottom of the figure, “Please indicate if you would consider health state A and B as better or worse than death.” The use of the word “and” at the bottom leads me to believe that the infant has all the conditions in both A and B. Is this because there are 2 options to answer at the bottom?

“The main limitation associated with this study is that no obvious state worse than death was found.” It seems that the descriptions “Severely disturbed sleep, severe feeding problems, severe breathing problems, severe stool problems, inconsolable crying, bleeding or cracked skin, low-energy/inactive/dull” do not describe the pain of the infant and so do not elicit a response from an adult that there is no obvious state worse than death. Is it possible that the adults who took this survey view the 4’s as problems that occur occasionally and do not see these issues as worse than death? Adults can forget that infants are altricial, making the 4’s much more severe than they are for adults. How do you expect the results would vary if you included an outcome statement after each option (e.g., “severe stool/poo problems, resulting in hospitalization”?

6. PLOS authors have the option to publish the peer review history of their article (what does this mean?). If published, this will include your full peer review and any attached files.

Reviewer #1: No

Reviewer #2: No

Reviewer #3: Yes: Julie Campbell

---

## [Author Response · Author response to Decision Letter 0]

3 Jan 2020

Reviewer #1

The purpose of this study is to generate an algorithm to estimate utilities for the infant quality of life instrument (IQI) using a two-step choice-based modelling approach. The approach and methods used for the study seem appropriate and I have just a few comments.

This study uses a population from the UK, USA and China-Hongkong and my question relates to why this population was used. Was it based on availability or was it to ensure generalisability? In addition, I am not entirely sure why the second study was limited to USA. Perhaps the authors need to comment on the generalisability of the results.

Thank you for pointing this out. We agree and clarify the choice for the population in the methods section now.

“These countries were selected for practical reasons, since they are culturally different yet share one language, thus eliminating the need for translation at this phase, and enabling the analysis of possible cross-cultural differences in the results, and therefore improving generalizability.”

It would help if a table with the utility values for the various health states is presented in the appendix. Linked to this point, the lowest/worst health state is represented by 0.015 and as a result, there is a gap between this value and zero which is usually considered the anchor point for utility scales. As a result of this, the use of the IQI would imply that there could potentially be an over-estimation of utility for the severest health states which could potentially lead to a situation where the paediatric population would be disadvantaged compared to e.g. an adult population. Perhaps, the authors should also comment on the implications of using this instrument to assess the cost-effectiveness of interventions.

Although we do agree that a complete table would be informative, we feel that presenting a complete list of all the estimated utilities for all IQI health states is not feasible, because we have in total 16,384 (47: 7 items with 4 levels) unique health states. Therefore, we have listed a few health states with their associated estimated utility values as an example (section 3.4). However, the complete list is available on request.

Indeed, the lowest utility for the IQI is 0.015. In the Discussion we address the lowest values for existing generic utility systems (e.g., EQ-5D, HUI, QWB, 15D, SF-6D, AQol-8D). Some of these instruments allow utility values to drop below 0.0 (worse than dead). The fact that the IQI has a lowest utility just above the value for dead, shows that overall respondents (caregivers and people from the general population) considered a baby in a very bad health condition to have a health state that is (slightly) better than death. To reflect on the possible consequences of this in cost-effectiveness analysis we added the sentence below in the Discussion.

“… since reimbursement decisions will be made based on a broader range of criteria than just cost-effectiveness, we do not expect that having the lowest IQI value higher than 0 would be a disadvantage for the infant population.”

Reviewer #2

General comments to the authors

Thank you to the authors for providing me with the opportunity to review this novel and interesting quality of life research paper. The authors identified a substantial gap in the literature in their previous published work and are now developing the algorithm for the previously established descriptive system of the IQI to capture and assess proxy-reported outcomes for the derivation of health state utility valuations for infants aged 0-1 years.

I note that the underpinning theory regarding the development of the IQI’s descriptive/classification system by Krabbe et al’s authorship team in 2018 (https://doi.org/10.1371/journal.pone.0203276) where the views of the primary caregivers serve as the proxy and provide the response. This paper is the logical next step to that published work.

Overall, I consider that it would be useful to provide some additional background in the Introduction section of the manuscript regarding the development of the IQI’s descriptive/classification system, including the literature search and use of expert panels in this development. This would contextualise the reasons for this paper to the broader readership.

Thank you for this suggestion to position our paper in our broader scientific work. Please see our reply on this issue on the next page (Specific comments to the authors).

I also suggest that the Discussion section could be more circumspect about the development of this initial (albeit important) value set. Please include caveats regarding future confirmatory and clinical studies and perhaps refinement of the value set over time.

Clinical studies using the IQI are currently going on. Their results will contribute to populate the data set on which the value set was built and refine it over time. We have added a sentence on this point in the discussion. 

Finally, I also suggest that the manuscript should be thoroughly checked for grammatical errors.

We have sent the manuscript to a professional language revision service before resubmitting it. 

I recommend the paper for publication if my key concerns are appropriately addressed as outlined below.

 

Specific comments to the authors

Introduction section

Line 69: The introductory paragraph could perhaps mention full health economic evaluation and cost-utility analyses to contextualise the generation of utilities as a health economic input metric for cost-utility analyses (refer Drummond et al 2015).

Thank you for this suggestion. We have inserted a reference to the handbook of Drummond et al. after the second sentence of the Introduction.

Line 86: As noted in the general comments, please include some additional background regarding the development of the IQI’s descriptive system.

Thank you for this suggestion. We have added the following text in the Introduction:

“For the selection of the relevant domains (items) for the IQI, a multi-step development process began by extracting candidate health concepts from relevant measures that were identified by searching the literature. Next, panels, with experts from Asia, Europe, New Zealand, and the United States of America (USA), and two surveys, with primary caregivers in New Zealand, Singapore, and the United Kingdom (UK), evaluated the relevance of the candidate health concepts, organized them into attributes based on their similarities, explored alternative attributes, and generated response scales. Additional interviews assessed the cross-cultural interpretability, parents’ understanding of health attributes, and the usability of the mobile application.”

Line 93 Aims: The ‘aims’ paragraph could be improved by describing how you planned to achieve your key objective with some further detail and perhaps numbered sequentially.

Thanks, for this suggestion. We have altered the aim paragraph into:

“The present study aimed to explain how utilities for the IQI health states were generated with a novel two-step choice-based modeling procedure that helps locate the position of death on the IQI scale [14]. Specifically, this process involved the following two steps: 1) deriving values for a set of IQI health states from primary caregivers of infants based on a discrete choice modeling exercise, 2) normalizing the values obtained in Step 1 to an anchored 0.0–1.0 scale using utilities derived from responses collected from a general population sample.”

 

Methods section

Line 108: please provide further explanation regarding the ‘skin’ dimension being classified ‘qualitatively’. Is the ‘mood’ dimension perhaps phrased qualitatively too? Do you mean that there needs to be a more subjective assessment of the response?

Indeed, as also explained in one of our previous publications [Jabrayilov et al., 2019] about the IQI valuation study, for mood and skin no monotonously decreasing coefficients were observed, confirming the qualitative (i.e., no logical ordering) nature of these two items. However, this that does not imply a more subjective assessment of the response, they can be analyzed just as the other items.

Jabrayilov R, Vermeulen KM, Detzel P, Dainelli L, van Asselt ADI, Krabbe PFM. The Infant health-related Quality of life Instrument (IQI): Valuing health status in the first year of life. Value Health 2019;22(6):721-727.]

Line 110: is the term ‘parents’ = primary carers? Are there ‘primary carers’ of the infants who are not classified as parents? Please clarify.

The two terms (parents and primary caregivers) were used alternatively because no caregiver who was not a parent has been interviewed. However, for reasons of consistency, we think that is more correct to stick to the same terminology in order to avoid confusion: the word parent has been replaced with caregiver.

Line 219: Could you please relabel this “Recruitment of Participants (or Respondents)”. Please expand on this fundamental section: the recruitment strategy requires additional explanation. I note that a fulsome explanation was provided of the discrete choice design and the subsequent analyses. It is insufficient to say that participants were reached through a market research company. I need to understand the sampling and any bias. I also need to understand why participants from the USA were recruited only for the second study. Please also clarify in this section the meaning of the ‘main study’ and the ‘additional study’.

We have rephrased the label of the subheading into “Recruitment of respondents”. In this section we also added more information about the sampling strategy and made clear what the purpose was of the main study and the additional study. We added the text below in the manuscript:

“These countries were selected for practical reasons, since they are culturally different yet share one language, thus eliminating the need for translation at this phase, and enabling the analysis of possible cross-cultural differences in the results, and therefore improving generalizability. Clear instructions were given to all participants. While the instrument targets infants up to one year, in the survey we chose to include primary caregivers of 2- and 3-year-olds as well, to enable the recruitment of a larger sample. We assumed that the caregivers could recollect their experiences of the first year of their infant’s life quite easily. The general population sample included both parents (of children with varying ages) and respondents without children, to be as representative as possible. The latter were interviewed because they might think differently about the value of life in different health conditions.

 In an additional study, a separate smaller sample was drawn in the USA from among members of the general population. This study was conducted to gain a better understanding of the severe IQI health states to explore and confirm our findings from the main study.”

Results section

Given the international nature of the cohort, stratified results should be reported beyond section 3.1 to highlight any cultural/country differences.

We have performed separate analyses for the three countries. In the new version of the paper, we have added a brief description of the results in the main text (lines 262-266, 299-304) and inserted 2 additional tables and 1 figure (S4 Table, S6 Table, S7 Fig) to present these.

Section 3.4: Please provide further contextualisation regarding the utility derived for ‘1111112’ and ‘1111111’. The reference point and the derivation of a utility score of 1.0 for 1111112 and 0.96 for 1111111 may require further clarification for the broader readership.

As stated in the methods section “After preliminary analyses to assess “interaction,” the reference category was changed to the second level, because the first level in this health item did not represent the best health condition; i.e., the second level had the highest coefficient. This implies that the first level “highly playful/highly interactive” would result in a lower score in the valuation than would the second level “playful/interactive.”

Then, in the results section where the utilities are derived, it says: ‘The best IQI state (utility score 1.000) was 1111112, since level 2 was the reference level for the interaction domain. State 1111111 had a utility score of 0.961’.

We assumed the matter was explained sufficiently in this way, but we propose to change the latter sentence into: ‘‘The best IQI state (utility score 1.000) was 1111112, since Level 2 was the reference level for the interaction domain, and therefore, 1111112 was considered to represent perfect health. State 1111111 had a utility score of 0.961”.

Discussion section

The Discussion is perhaps the weakest section of the paper and should be expanded to showcase the key and secondary findings of the paper, thoroughly outline all of the limitations of the study, and provide a robust conclusion. The first paragraph should provide an overall summary of the findings. One sentence is inadequate and this sentence describes what was done rather than a succinct summary of the key findings of the paper.

Thank you for this suggestion. We have rewritten the first paragraph, see below:

“In this study, we explained a novel two-step procedure to generate utilities for the IQI, relying on a sample comprising both primary caregivers and members of the general population. Values were derived for IQI health states based on responses from primary caregivers (e.g., proxies). Results indicated that out of the 7 IQI items, “breathing” had the highest impact on the HRQoL of infants. Moreover, except for “stooling,” all item levels were statistically significant. Subsequently, these values of the caregivers were normalized into utilities by using information on the location of death on the scale, derived from a general population sample. Findings revealed that none of the health states contained in the IQI was worse than death.”

The Discussion also does not outline the international nature of the cohort, nor issues surrounding cultural differences for the proxy-respondents as primary carers of infants and therefore the generation of potentially different value sets?

We have now added results per country in supplementary tables and reported about differences between countries in the main text of the results section. We also reflected on the cultural differences between countries in the discussion section. We do believe that eventually, country-specific value sets are warranted, but the primary aim of this study was to demonstrate the feasibility of the method, and not to generate local value sets. 

Added to discussion: 

“The country-specific results for the primary caregivers revealed a number of differences. Overall, the UK and USA appeared more alike, while China was slightly different. This could have its source in the different cultures of the countries in the sample, but given that sample sizes of around 400 were used for each country, a part of this finding may also be an artefact. In China, sleep appeared to be more important, whereas in the UK and USA, higher coefficients were observed for most other attributes, and more value was attached to breathing, mood, and interaction. Eventually, a larger sample should be used to determine final value sets based on country-specific preferences.” 

In the second paragraph of the Discussion, perhaps remove the word ‘first’ – the authors do not then go to ‘second’, ‘third’ etc to expand on the key findings. The second paragraph could also reference the AQoL-8D multi-attribute utility instrument’s algorithmic range as an example of an instrument that does not record a utility value that is less than zero.

Thank you for noticing this. We have dropped the word “First”. In addition, we have inserted a new reference to the Australian AQol-8D instrument [Richardson, 2014]. In this study, based on another study sample than the study presented in Culyer’s Encyclopedia, Figure 3 is indeed showing that the lowest AQol-8D health states is approximately 0.25.

Richardson, J, Sinha K, Iezzi A, Khan, MA. Modelling utility weights for the Assessment of Quality of Life (AQoL)-8D. Quality of Life Research. 2014;23:2395-2404.

 

In the third paragraph, I would remove the statement that “It is likely that states worse than death are less self-evident than generally thought and that the lowest utility for the EQ-5D-3L may have been an accidental finding”. Perhaps the suggestion could be underpinned by statements about the instruments only 243 health states and that the instrument is relatively insensitive to complex and chronic disease states. Please use appropriate referencing for this statement.

We have dropped this sentence.

The fourth paragraph could be tightened to provide some additional explanation and contextualisation regarding the underpinning model and the advantages of the model. This paragraph in its current form is somewhat jumbled.

Thank you for this suggestion. We have rewritten this paragraph (Discussion: lines 351-357).

Limitations section. Please be more circumspect in this section – one limitation only is outlined. The recruitment/sampling strategy is not properly described therefore I cannot comment on limitations regarding the sampling strategy. Similarly, stratified results are not presented I would expect that cultural differences would be evident in stratified results. Stratified results could also be the subject of further discussion.

Another limitation was that detailed characteristics of the respondents were not available, apart from country, age and sex. Therefore, we cannot say for instance which part of the general population sample were also parents, or perform stratified analyses based on socioeconomic status. The stratification of the results per country indeed revealed a number of differences. However, it should be kept in mind that the present study was largely intended as a proof of principle to demonstrate how the process of generating normalized values (utilities) can take place and to provide a first value set for the IQI. From that perspective, the generalizability of the results is of less importance at this stage. We added sentences about this in the Discussion (lines 371-377).

“Another limitation was that detailed characteristics of the respondents were not available, apart from country, age, and sex. Therefore, we were not able to, for instance, say which part of the general population sample comprised parents, nor could we perform stratified analyses based on socioeconomic status. It should be kept in mind that the present study was largely intended as a proof of principle to demonstrate how the process of generating normalized values (utilities) can take place and to provide a first value set for the IQI. From that perspective, the generalizability of the results is of less importance at this stage.”

 

The conclusion should provide additional explanation regarding future research and confirmatory studies.

Clinical studies using the IQI are currently going on. In some of those, another health instrument is used in parallel, in order to double check the IQI validity (e.g. results going in the same direction: subjects improving their health states confirmed by both instruments). Moreover, since the IQI is a “living” instrument, these results will contribute to populate the data set on which the value set was built and refine it over time. We have added a sentence on this point in the discussion. 

Reviewer #3

The sample consisted of members of the general population (n=1409) and infant caregivers (n=1229). Were the members of the general population parents? If not, is there reason to believe that the sample of the general population would answer the questions significantly differently than the infant caregivers?

The general population sample did contain parents, as we aimed it to be representative of the general population. And yes, we feel there is reason to believe that this sample may have different preferences than the primary caregivers. We added the following to the methods section (2.2 recruitment):

“The general population sample included both parents (of children with varying ages) and respondents without children, to be as representative as possible. The latter were interviewed because they might think differently about the value of life in different health conditions.”

I’m assuming from some of the information that follows in the manuscript, that yes, the subsamples answer some questions differently, but in section 3.2 “states worse than death”, the authors write, “Among the health states in the main study, the one most frequently mentioned as being worse than death was 4241241 (2.2%) for the general population and 4244231 (2.4%) for the primary caregivers.”

We indeed assumed there would be differences, also given previous (from literature) difficulties with estimating utilities for children as parents were not willing to trade any life years no matter how bad the health state. In the end, as seen by our data, they turn out not be so different, which is still an interesting finding to report.

The digits are poorly explained in section 2.1 instrument. A slightly more detailed explanation would be useful for the reader earlier in the paper (perhaps expand the example given in line 113, “e.g., 3231421 equates to moderately affected sleep, slight feeding problems, moderate breathing problems, sleeps well, inconsolable crying, dry or red skin, highly playful/highly interactive.” Because as it is, I’m not even certain that I have interpreted this correctly. Is this correct?

We agree to the reviewer’s suggestion and have expanded the example to include the explanation of the levels (which was indeed correctly listed by the reviewer, apart from the 4th attribute which was not ‘sleeps well’ but ‘normal stool/poo’ as sleeping is the 1st attribute), as follows: “… e.g., 3231421, which would equate “moderately affected sleep, slight feeding problems, moderate breathing problems, normal stool/poo, inconsolable crying, dry or red skin, highly playful/highly interactive”. 

 

Figure 2 is confusing because at the top it says, “Suppose that an infant’s first year of life is spent mainly in either State A or State B and that its health is uncertain afterwards.” At the bottom of the figure, “Please indicate if you would consider health state A and B as better or worse than death.” The use of the word “and” at the bottom leads me to believe that the infant has all the conditions in both A and B. Is this because there are 2 options to answer at the bottom?

The health status A and B are two different alternatives, as stressed by EITHER and OR in the initial sentence. An infant might have experienced one OR the other, not both of them together. After this first question, no matter which health status the infant has experienced (A or B, we do not know and it does not matter), in the second question (note that these are two separate/independent questions) we are asking to evaluate whether A AND B, each of them separately, are better or worse than death according to the interviewed person. Given results from the survey, most respondents seemed to have understood this well.

 

“The main limitation associated with this study is that no obvious state worse than death was found.” It seems that the descriptions “Severely disturbed sleep, severe feeding problems, severe breathing problems, severe stool problems, inconsolable crying, bleeding or cracked skin, low-energy/inactive/dull” do not describe the pain of the infant and so do not elicit a response from an adult that there is no obvious state worse than death. Is it possible that the adults who took this survey view the 4’s as problems that occur occasionally and do not see these issues as worse than death? Adults can forget that infants are altricial, making the 4’s much more severe than they are for adults.

This is a rather academic discussion: many of these thoughts are also applicable to generic utility instruments, which rarely specify a time-period when asking to evaluate a health state. The interviewed person can therefore imagine anything about the length of that condition. In the same way, we do not know how the adults have interpreted the survey. What we do know (as it was specified in the question) is that these infants are experiencing that specific health condition in the first year of life and what happens later in time is uncertain.

How do you expect the results would vary if you included an outcome statement after each option (e.g., “severe stool/poo problems, resulting in hospitalization”?

Our way of reasoning was that, by describing a severe health state of an infant, caregivers would realize the potential implications of such a health state. Adding more information about the consequences, such as hospitalization or not, might indeed affect the preferences for one health state over another, but this was not the purpose of the exercise.

---

## [Decision Letter · Decision Letter 1]

11 Mar 2020

A two-step procedure to generate utilities for the Infant health-related Quality of life Instrument (IQI)

PONE-D-19-19703R1

Dear Dr. Krabbe,

We are pleased to inform you that your manuscript has been judged scientifically suitable for publication and will be formally accepted for publication once it complies with all outstanding technical requirements.

With kind regards,

Jing Tian

Academic Editor

PLOS ONE

Additional Editor Comments (optional):

Reviewers' comments:

Reviewer's Responses to Questions

**Comments to the Author**

1. If the authors have adequately addressed your comments raised in a previous round of review and you feel that this manuscript is now acceptable for publication, you may indicate that here to bypass the “Comments to the Author” section, enter your conflict of interest statement in the “Confidential to Editor” section, and submit your "Accept" recommendation.

Reviewer #2: All comments have been addressed

Reviewer #3: All comments have been addressed

2. Is the manuscript technically sound, and do the data support the conclusions?

Reviewer #2: Yes

Reviewer #3: Yes

3. Has the statistical analysis been performed appropriately and rigorously? 

Reviewer #2: Yes

Reviewer #3: Yes

4. Have the authors made all data underlying the findings in their manuscript fully available?

Reviewer #2: Yes

Reviewer #3: Yes

5. Is the manuscript presented in an intelligible fashion and written in standard English?

Reviewer #2: Yes

Reviewer #3: Yes

6. Review Comments to the Author

Reviewer #2: Dear authors, thank you for responding to all of my comments and suggestions in a thorough and robust manner. I now recommend the article for publication. Good luck with your publication and future endeavours with this interesting work. With kindest regards.

Reviewer #3: The authors were careful to respond to all my inquiries and addressed them in the manuscript when necessary for clarification.

7. PLOS authors have the option to publish the peer review history of their article (what does this mean?). If published, this will include your full peer review and any attached files.

Reviewer #2: Yes: Dr Julie A. Campbell

Reviewer #3: Yes: Julie Campbell